# Black Tea Waste as Green Adsorbent for Nitrate Removal from Aqueous Solutions

**DOI:** 10.3390/ma16124285

**Published:** 2023-06-09

**Authors:** Andreea Bondarev, Daniela Roxana Popovici, Cătalina Călin, Sonia Mihai, Elena-Emilia Sȋrbu, Rami Doukeh

**Affiliations:** 1Chemistry Department, Petroleum-Gas University of Ploiesti, 39 Bucharest Blvd., 100680 Ploieşti, Romania; andreeabondarev@yahoo.com (A.B.); dana_popovici@upg-ploiesti.ro (D.R.P.); oprescuemilia@gmail.com (E.-E.S.); 2National Institute for Research & Development in Chemistry and Petrochemistry ICECHIM, 060021 Bucharest, Romania

**Keywords:** nitrate, adsorption, tea waste, pollution

## Abstract

The aim of the study was to prepare effective low-cost green adsorbents based on spent black tea leaves for the removal of nitrate ions from aqueous solutions. These adsorbents were obtained either by thermally treating spent tea to produce biochar (UBT-TT), or by employing the untreated tea waste (UBT) to obtain convenient bio-sorbents. The adsorbents were characterized before and after adsorption by Scanning Electron Microscopy (SEM), Energy Dispersed X-ray analysis (EDX), Infrared Spectroscopy (FTIR), and Thermal Gravimetric Analysis (TGA). The experimental conditions, such as pH, temperature, and nitrate ions concentration were studied to evaluate the interaction of nitrates with adsorbents and the potential of the adsorbents for the nitrate removal from synthetic solutions. The Langmuir, Freundlich and Temkin isotherms were applied to derive the adsorption parameters based on the obtained data. The maximum adsorption intakes for UBT and UBT-TT were 59.44 mg/g and 61.425 mg/g, respectively. The data obtained from this study were best fitted to the Freundlich adsorption isotherm applied to equilibrium (the values R^2^ = 0.9431 for UBT and R^2^ = 0.9414 for UBT-TT), this assuming the multi-layer adsorption onto a surface with a finite number of sites. The Freundlich isotherm model could explain the adsorption mechanism. These results indicated that UBT and UBT-TT could serve as novel biowaste and low-cost materials for the removal of nitrate ions from aqueous solutions.

## 1. Introduction

Innovative environmentally friendly methods to recycle industrial and food waste are needed at present because of the recent acceleration of climate change. The world is increasingly heading toward a closed-circuit economy, an alternative economic model centered on the idea of ending the life cycle of a commodity [1,2,3,4].

Recent findings from laboratory research indicate that it may be possible to develop effective and reasonably priced adsorbents using various by-products from the agro-food industry, such as tea waste, coffee waste, or seed and fruit waste, to remove organic compounds and heavy metals from contaminated aquatic environments. An abundant natural substance or one that is a waste product or byproduct of industry or agriculture that requires little to no processing is considered a low-cost adsorbent [5,6,7,8,9,10].

Following the “3 Rs”, the circular economy seeks to address pollution issues by returning of materials to another life cycle. In such a situation, waste is seen as a value-added substance [7,10,11].

Due to their high solubility and nutritional importance for aquatic plants, nitrates are important components of agricultural fertilizers. In order to meet their nutrient needs, plants use nitrates from the soil, and they can build up nitrates in their leaves and stems [12,13]. Nitrates used in chemical fertilizers are highly soluble in water and do not effectively bind to soil particles [14,15].

A high level of nitrate in drinking water is associated with many health problems, including the development of thyroid diseases, infectious diseases, colon cancer risk, and blue baby syndrome in newborns (methemoglobinemia). The maximum contaminant limit (MCL) for nitrates in drinking water is 10 mg/L nitrate-nitrogen or 45 mg/L when the concentration is expressed as a nitrate, according to the World Health Organization (WHO). Increased levels of nitrate in drinking water have been linked to several health issues, including the development of thyroid diseases, infectious diseases, colon cancer risk, and blue baby syndrome in newborns (methemoglobinemia). Pregnant women can pass methemoglobin on to developing fetuses, and low birth weights have been attributed to high nitrate content in water. High levels of nitrate can pose a risk to adults with specific rare metabolic disorders. This can affect breathing and be life-threatening in severe cases [16,17].

Hence, from the perspective of protecting both human health and the environment, nitrate ion removal has significantly increased in importance.

Recent research has shown that the low anion exchange capacity of soils frequently limits the removal performance of nitrate. As a result, carbon additions such as woodchips, sawdust, crop residues and biochar have been suggested for nitrate removal [18,19].

A new category of adsorbents named “metal-organic frameworks” (MOFs) was presented in the recent literature. MOFs have been widely used as photocatalysts, supercapacitors, in bio-medical imaging, and as adsorbents for gases. New research studies started to investigate MOFs as adsorbents in wastewater or natural water, owing to their surface area and porosity. Recent studies presented the elimination of Rhodamine B toxic dye, one of the usual industrial pollutants of natural water, by using a novel bimetallic zeolitic imidazolate framework (ZIF) which was fabricated by a simple and straightforward technique at room temperature [20].

In the available literature, it has been presented that hexaferrite materials (Yb-Zn doped CaPbFe_12_O_19_ M type hexaferrites) are excellent materials for the removal of textile contamination. M type hexaferrites are promising photocatalyst materials, and possess a band gap in the range of 1.1–2.3 eV. These materials have an excellent photocatalytic activity towards the degradation of methylene blue dye under sunlight in synthetic water [21].

Arsenic recovery from acidic wastewater and acid recycling are very important for the metallurgical industry with regard to reducing pollution due to carbon emissions. Recent research studies presented the possibility of arsenic recovery by an adsorption process using a hydroxyl-enriched CeO_2_ adsorbent. Current recovery methods of some chemical elements are also concentrated on chemical leaching methods. For example, recovering Li from spent batteries is possible by a direct electro-oxidation leaching approach, and the pollution problem is reduced [22,23].

The adsorption method is a simple, easy-to-use, and scalable method of contaminant removal [19,24]. For the removal of nitrate ions from aqueous solutions, a variety of adsorbents have been used, including hybrid nanocomposites, activated carbon, anionic layer double hydroxide, and magnetic nanoparticles [6,25].

Different plant waste biomass has been converted into activated carbon and has been used to remove different contaminants from wastewaters [25,26]. However, these materials are expensive and can achieve high surface areas after carbonization and activation. Due to the high production cost of activated carbon and the challenges of the regeneration step, activated carbon is considered a standard efficient adsorbent for the removal of various types of contaminants in aqueous environments. Therefore, the development of affordable and environmentally friendly adsorbents has become an important research topic in recent years [26,27,28,29].

The biosorption of pollutants from wastewater is now frequently carried out using green adsorbents, a novel class of biosorbents made from agricultural waste. These wastes are gradually replacing the commercially available adsorbents because they have several key advantages, including low cost, availability, the ability to be recycled into products with added value for waste removal, a rich chemical composition with high surface area, and the potential for modification. The literature reveals a growing demand for biosorbents that are environmentally friendly and which have low costs [29,30].

One of the inexpensive green adsorbent plant materials for the elimination of various contaminants is tea. Tea leaf cellulose and hemicelluloses, lignin, structural proteins, and condensed tannins make up most of the cell wall [18,31].

Due to the variety of functional groups contained in tea leaves, they have excellent potential for the removal of organic or inorganic contaminants from wastewater. Large amounts of solid tea waste could be used to improve soil fertility [9,13]. The second most popular beverage after water is tea. According to a study by the United Nations Food and Agricultural Organization (FAO), global exports are rising and are projected to reach 750,981 tons by 2023 [3,31].

In the current study, black tea waste was evaluated as a potential biosorbent for the removal of nitrate ions from synthetic wastewater samples. The adsorbents were obtained either by thermally treating spent tea to produce biochar (UBT-TT), or by employing the untreated tea waste (UBT) to obtain convenient bio-sorbents.

The adsorbents were characterized before and after nitrate adsorption by Scanning Electron Microscopy (SEM), Energy Dispersed X-ray analysis (EDX), Infrared Spectroscopy (FTIR) and Thermal Gravimetric Analysis (TGA). The experimental conditions, such as pH, temperature and nitrate ions concentration, were studied to evaluate the interaction of the nitrate with adsorbents and the potential of the adsorbents for the nitrate removal from synthetic solutions. The Langmuir, Freundlich and Temkin isotherms were applied to derive the adsorption parameters based on the obtained data.

The reusability potential of the adsorbents was assessed to elucidate their efficacy.

## 2. Materials and Methods

### 2.1. Materials and Reagents

All chemicals and reagents used in this study were of analytical grade and were utilized as supplied without any additional treatment.

Potassium nitrate (KNO_3,_ analytical reagent, ≥98.0%, Merck, Darmstadt, Germany) was used to prepare the model (synthetic) solutions. A stock solution of a concentration of 1000 mg L^−1^ (as N-NO_3_^−^) was prepared by dissolving 1.6290 g KNO_3_ in 1000 mL demineralized water. The experimental solutions were prepared by diluting the stock solution to the desired concentration (50–500 mg L^−1^). Used black tea adsorbents were prepared from commercial fresh black tea leaves.

Hydrochloric acid (HCl, 36~38%, analytical reagent, ≥99.0%), sodium hydroxide (NaOH, ≥98.0%), sodium carbonate (Na_2_CO_3,_ analytical reagent, ≥98.0%) and salicylic acid (C_7_H_6_O_3,_ ≥99.0%) were purchased from Merck Millipore, Darmstadt, Germany.

### 2.2. Preparation of Adsorbent

Used black tea adsorbent (UBT) was prepared from commercial fresh black tea leaves by boiling with distilled water for 20 min and then filtering. UBT was boiled repeatedly until the remaining solution became colorless. This treatment was repeated eight times. Washing should remove all substances contained in spent tea that could cause contamination (such as polysaccharides, tannins, and colored components). After the extraction of black tea liquor, the leaves were dried in an oven (Binder, Germany) at 105 °C for 48 h. Dried tea residues were ground and sieved through the sieves of mesh sizes ≤ 0.5 mm, and they were stored in air-tight bottles for adsorption experiments. These adsorbents were labeled ‘used black tea’ (UBT).

Portions of these samples were further thermally treated at 400 °C for 2 h, and the resultants were labeled ‘thermally treated black tea waste’ (UBT-TT). The carbonaceous material was carbonized at 400 °C in a muffle furnace (Nabertherm, Germany). The final biochar obtained was washed thoroughly after cooling with distilled water until the pH of the rinsing water was neutral. The adsorbent thus obtained was dried in an oven at 105 °C for 24 h, and then sieved with an 80–200 mesh sieve.

Finally, the resulting adsorbents were used for the experimental adsorption studies.

### 2.3. Preparation of the Samples Artificially Contaminated with NO_3_^−^ Ions

Water samples artificially contaminated with NO_3_^−^ ions were prepared by dissolving the required amounts of KNO_3_ in deionized water to a final concentration of 500 mg/L stock solution. Further dilutions were prepared in the same solvent.

The pH of the solutions was adjusted to the desired value using 0.1 mol/L HCl or NaOH solutions. The calibration curve for the determination of NO_3_^−^ at the selected pH value was prepared by measuring different concentrations of NO_3_^−^ (50–500 mg/L) at 410 nm.

The concentration ranges existing in the industrial raw effluents was selected for this research, and after the variation of the initial pollutant concentration, the value 300 mg/L was chosen for the working concentration. The desired concentrations were obtained by a dilution technique. All other chemicals used in the present study were of laboratory grade, and standard procedures were followed for all experiments.

### 2.4. Adsorbent Characterization

UBT and UBT-TT morphology and surface characteristics were studied using a field emission scanning electron microscope coupled with energy-dispersive spectra EDS (FEI Scios DualBeam FIB SEM, Thermo-Fisher, Brno, Czech Republic). The Fourier transform infrared spectrometer (FT-IR) (Shimadzu IR TRACER-100, Kyoto, Japan) was used for the identification of functional groups on the UBT and UBT-TT surface involved in nitrate adsorption.

The thermal decomposition characteristics of UBT and UBT-TT adsorbents were monitored using TGA/DTG (thermogravimetric/derivative thermogravimetric).

### 2.5. Adsorption Equilibrium Studies

To evaluate the nitrate adsorption performance of used black tea (UBT) adsorbents, batch experiments were carried out in a batch-type procedure using a shaker (benchtop shaker laboratory incubator 300 × 400 MM, 70DEG C, Essex, UK). Samples in Erlenmeyer flasks were removed from the shaker at predetermined time intervals. After filtration, the residual nitrate concentration was determined at 410 nm using a UV/Vis spectrophotometer (CECIL 1021, 1000 Series, Bristol, UK) according to salicylic acid method [32,33].

The influence of the nitrate concentration (50–300 mg/L), pH (2.0–10.0), and temperature (15–40 °C) on the adsorption process were studied. The initial solution pH value was adjusted by adding 0.1 mol/L HCl or NaOH solutions. For all experiments, the adsorbent weight was 0.2 g and the contact time was 90 min.

The kinetics were examined at various nitrate concentrations, and the isotherms were carried out at different temperatures. The amount of nitrate retained by the adsorbents, *q_e_* (mg g^−1^) for all samples was calculated by Equation (1) [12,13,14,15,29]:
(1)$$q_{e}\;=\;(C_{0}-C_{e})\;\frac{V}{W}$$
where *q_e_* is the nitrate uptake (mg g^−1^); *C*_0_ (mg/L) is the initial concentration of the nitrate; *C_e_* (mg/L) is the equilibrium nitrate concentration, at equilibrium; *V* is the volume of the solution (L), and *W* is the weight of the adsorbent (g).

## 3. Results and Discussion

### 3.1. Characterization of the Adsorbents

#### 3.1.1. SEM Analysis

To determine the structure and morphology of the adsorbent materials, Scanning Electron Microscopy (SEM—FEI Scios DualBeam FIB SEM) coupled with Energy-Dispersive Spectra (EDS), equipped with a field emission gun and a 1.2 nm resolution X-ray energy dispersive spectrometer with a resolution of 133 eV were used (Thermo Fischer). The FEI Scios™ is an ultra-high resolution DualBeam™ analytical system that achieves outstanding 2D imaging for a wide range of samples.

The original adsorbent, which was waste black tea (UBT), had a diversified, coarse and highly porous surface composed of numerous fibrous bonds and spongy formations, providing a wide field for the adsorption of the nitrate (Figure 1a,b). The main components of tea are cellulose and hemicellulose, and it was observed that UBT presented a stem structure (Figure 1a).

A rougher surface area and the broadly distributed pores of UBT-TT can offer an effective surface area and more opportunities for the binding of nitrate ions. The UBT-TT-loaded adsorbent showed uniform coverage of UBT-TT by nitrate, leading to a reduction in surface porosity. This surface characteristic would substantiate the higher adsorption capacity of UBT-TT (Figure 1c,d).

##### Energy Dispersed X-ray (EDX) Measurements

The EDX spectra of adsorbents (UBT and UBT-TT) are shown in Figure 2. The EDX spectra of adsorbent sample UBT showed that on the surface of all the samples, carbon and oxygen were mostly present. The EDX spectra also showed the presence of other elements, such as Mg, K, Mn, Si, P, S and Mn (Figure 2a). After the thermic treatment, the content of carbon increased from 54.13 to 63.06%, while the oxygen decreased from 45.05 to 36.59% due to the loss of moisture and some phosphate and sulfate groups. This affirmation is supported by the decrease in the values recorded for P and S from 0.20 to 0.13%, and 0.18 to 0.1% (Figure 2b), respectively. This behavior was also reported by P. Wijeyawardana et al. [34] and S. Suman et al. [35].

#### 3.1.2. ATR-FTIR Analysis

Attenuated Total Reflectance Fourier Transform Infrared Spectroscopy (ATR-FTIR) was used to characterize the samples before and after the addition of NO_3_^−^. The surface chemistry of UBT and UBT-TT was studied using a FTIR Spectrophotometer (Shimadzu IR TRACER-100, Kyoto, Japan) in the region of 4000–400 cm^−1^. The spectra of UBT and UBT-TT samples before and after nitrate adsorption were examined and compared to identify the functional groups on the adsorbent (Figure 2, Figure 3 and Figure 4). Like other biomasses, black tea leaves primarily consist of cell walls (comprised of cellulose, hemicellulose, lignin, proteins, condensed tannins, etc.). These components have a variety of functions (hydroxyl, carboxylate, etc.) that can effectively help remove various contaminants [8].

To study the effect of thermal treatment on biochar structures, the FT-IR analysis was used to identify the functional groups from UBT and UBT-TT. As can be seen in Figure 3, the UBT sample exhibits a broad band around 3310 cm^−1^ and a peak at 1624 cm^−1^, which is attributed to the stretching vibration of the banded –OH groups from alcohols [15]. The peaks at 2358, 2918 and 2848 cm^−1^ are attributed to C≡N stretch [36] and the C–H and −CH_2_- stretching vibration modes [1]. The absorption peaks in the region of 1458–837 cm^−1^ indicates the presence of esters and ethers groups [37].

After thermal treatment, the corresponding absorption peaks to esters, ether groups, and C–H and −CH_2_- stretching vibration modes decrease significantly in terms of intensity. The UBT-TT exhibit a broad peak at 3313 cm^−1^ attributed groups from alcohols, [15] and two peaks at 2358 and 1595 cm^−1^ assigned to the C≡N stretch [36] and the amine group [38].

The FT-IR spectra of tea biomass before and after nitrate loading were very similar (Figure 4). The only significant difference was the decrease in peak absorbance at 3310, 1624 and 1031 cm^−1^ that indicated that the -OH and C-O groups were involved in nitrate adsorption [28].

The same behavior was also observed for thermal treated tea biomass (UBT-TT) before and after nitrate adsorption (Figure 5). However, a slight difference was observed consisting of the splitting of the band at 1595 cm^−1^ into two small sharp bands at 1562 and 1543 cm^−1^, probably due to the N-H bending vibration of the secondary amide and the primary amine [39].

### 3.2. Thermogravimetric Analysis: Heat Treatment of Bio-Sorbents and the Adsorption Performance

The extent of heating and carbonization was evidenced from the thermo-gravimetric analysis (Figure 6, Figure 7 and Figure 8). Since the thermal treated adsorbents have already undergone a thermal decomposition process in the production step, the weight loss of the bio-sorbent in TGA testing was not particularly large.

The thermal decomposition curves of UTB and UTB-TT are presented in Figure 6, Figure 7 and Figure 8. As can be seen for both samples, the first degradation step with a weight loss of 6.45% for UTB and 8.84%, for UTB-TT, respectively, occurred between 40–130 °C due to the elimination of moisture and volatile compounds [3]. The UTB biomass exhibits a maximum weight loss of 58.53% around 352 °C that is caused by the pyrolysis of the cellulose and hemicellulose content [37]. The slow weight-loss process of lignin pyrolysis took place for UTB in the region from 520 to 700 °C (Figure 6). Compared with UTB, the UTB-TT exhibits only the degradation step of the lignin residue [40] at 472 °C with a weight loss of 66.78% because of the thermal treatment at 400 °C, which caused the degradation of cellulose and hemicellulose. This observation is in agreement with FT-IR data.

The TGA curves of UTB and UTB-TT after nitrate adsorption (Figure 7 and Figure 8) had similar degradation behavior, with the mention of the presence of an additional decomposition step of 9.25% around the temperature of 200 °C for UTB and 9.53% at 240 °C for UTB-TT, which can be caused by the degradation of the absorbed nitrate anion.

### 3.3. Adsorption Performance

Numerous factors, such as initial adsorbate concentration, adsorbent dose, pH and temperature can affect the adsorption process differently. The adsorbent amount was 0.2 g and the contact time was 90 min.

#### 3.3.1. Effect of the Initial Concentration

The adsorption capacity of UBT and UBT-TT samples, as a function of the initial concentration of a nitrate in aqueous solution is presented in Figure 9a,b The adsorbent sample was added to 50 mL of the nitrate solution. The influence of the initial nitrate concentration (50–300 mg/L) was studied.

When the nitrate concentration was low, there were more adsorption sites on the surface of the adsorbent, and the adsorption efficiency of the nitrate was higher. However, the amount of nitrate exceeded the saturated adsorption sites of the adsorbent when the initial concentration of nitrate was higher, and the removal efficiency decreased gradually. As a result, increasing the initial nitrate concentration reduced the removal efficiency of the adsorbent due to the limited number of adsorption sites. By increasing the initial concentrations of nitrite, the specific sites of the adsorbents become saturated, and the exchange sites fill up. Although the kinetics of the process are slowed by higher initial nitrate concentrations, the uptake percentage increases [6,19].

#### 3.3.2. Effect of Initial pH

In this study, 0.2 g of adsorbent was added to 50 mL of nitrate solution. The influence of the initial pH was studied in the domain of 2.0–10.0. The initial solution pH value was regulated by 0.1 mol/L HCl or NaOH solutions.

Solution pH has an important effect on nitrate adsorption. Most adsorbents lose their adsorption capacity under strongly acidic or basic conditions. The change in adsorption with pH (Figure 9) can be explained by electrostatic interactions between adsorbents and adsorbates. The lowest values of the adsorption capacity were obtained at pH 2 and 4 for UBT and UBT-TT. Adjusting the pH with HCl affects the adsorption capacity. At lower pH values, the H+ ions compete with nitrate ions for the electrostatic surface charges in the system, decreasing the percentage of sorption [41]. The best values of the adsorption capacity were obtained at pH 6.5 for UBT and pH 8 for UBT-TT (Figure 10a,b). The highest adsorption capacity was achieved at neutral pH, mainly due to strong electrostatic interactions between the adsorption sites and the nitrate anions. A lower adsorption of nitrate at alkaline pH may be due to abundant HO- ions competing with contaminants for the same sorption sites.

#### 3.3.3. Effect of Temperature

The adsorption capacity of nitrate anions reached the highest values at 15 °C for UBT-TT and at 25 °C for UBT (Figure 11a,b). Temperature is a factor that influences the mobility of nitrate ions in aqueous solutions as well as the surface properties of the adsorbent. The increased adsorption at higher temperatures is caused by the adsorption sites being more readily available [15,29].

### 3.4. Regeneration Experiments

The effect of the type of the desorbing agent should be evaluated while taking into account the sorbent’s potential for reuse. Each adsorbent must have good reusability and high adsorption capacity to significantly increase the economic value of the process. Due to the long drying time, physical regeneration methods such as water regeneration and heat regeneration are less effective, less stable, and therefore more costly. The adsorbent is frequently regenerated using straightforward chemical reagents such as acid, alkali, and salts, because the chemical regeneration method offers the advantages of high regeneration efficiency and low cost [29,42].

For the regeneration of the adsorbents, several attempts were performed to regenerate nitrate ions from the adsorbed support by using HCl, NaOH and Na_2_CO_3_ solutions at the concentration of 0.1 mol/L. Among them, Na_2_CO_3_ and HCl solutions had the best regeneration effect due to the enhancement of electrostatic repulsion. After the sorption process, the dried and weighed sorbents (with an initial solution concentration of 300 mg/L) were stirred for 120 min with the desorbing agents HCl, NaOH or Na_2_CO_3_ at a concentration of 0.1 mol/L. After completion of the stirring, the adsorbent was separated by filtration and was then dried at room temperature for 24 h.

The desorption percentage was calculated using the expression [18,19,42]:$$\eta \;\%=\frac{C_{r}\;\cdot \;V_{t}}{\left(Co-Ce\right)\cdot V}\cdot 100$$
where: *C*_o_ (mg/L) is the initial concentration of NO_3_^−^ ions in the feed solution, *C*_e_ (mg/L) is the NO_3_^−^ concentration at equilibrium, *V* (L) is the volume of feed solution, *C_r_* (mg/L) is the NO_3_^−^ concentration in solution after regeneration, and *V_t_* (L) is the volume of the regeneration solution.

Figure 12a–c showed the repeatability of nitrate adsorption by each adsorbent after 0.1 mol/L HCl, NaOH, and Na_2_CO_3_ treatment. The results showed that the third cycle was almost 100% for all regeneration reagents, which demonstrated that each adsorbent prepared from spent black tea had good reusability.

### 3.5. Isotherm Analysis

In this study, the relationship between adsorption capacity and equilibrium concentration was analyzed by using the isotherm models of Langmuir, Freundlich and Temkin to fit the experimental data (Table 1).

The design and optimization of an adsorption process for the removal of inorganic and organic contaminants from aqueous solutions critically depends on the analysis of the equilibrium isotherms (Figure 13, Figure 14 and Figure 15).

The Langmuir parameters can also be used to predict the affinity between the nitrate and each sorbent using a dimensionless separation factor (*R*_L_) [3,10,31]:(2)$$R_{L}=\frac{1}{1+K_{L}\cdot C_{i}}$$
where *K_L_* is the Langmuir constant, and *C_i_* is the initial concentration (mg/L). 

According to the criteria listed in Table 2, the value of the separation factor *R*_L_ can be used to determine whether the adsorption process was favorable or unfavorable.

The value of *R*_L_ for the adsorption of nitrate onto UBT and UBT-TT adsorbents is shown in Figure 16, which indicates that the sorption of nitrate ions was favorable (see Table 2). The *R*_L_ value was between 0 and 1, which signifies a strong relation for adsorption (Figure 16).

The intensity of the adsorption process is indicated by the values of the Freundlich parameter *n*_F_. A favorable adsorption process is represented by *n*_F_ values between 1 and 10. The results showed that the value of 1/*n*_F_ is less than the unity, indicating that the nitrate pollutant is favorably adsorbed by the adsorbents prepared from black tea waste. This is in strong agreement with the findings regarding the *R*_L_ values [2,15,31].

The data obtained from this study were best fitted to the Freundlich adsorption isotherm applied to the equilibrium (the values were R^2^ = 0.9431 for UBT and R^2^ = 0.9414 for UBT-TT), assuming the multi-layer adsorption onto a surface with a finite number of sites [10,31].

Since the Langmuir and Freundlich isotherm models did not sufficiently explain the adsorption mechanism, the Temkin isotherms were also fitted to the experimental data. The adsorption parameters calculated using the Temkin model indicated that the electrostatic interactions participated in the nitrate sorption process.

The data obtained from this study (Table 3) were best fitted to the Freundlich adsorption isotherm applied to the equilibrium (the values R^2^ = 0.9136 for UBT and R^2^ = 0.9506 for UBT-TT), assuming the multi-layer adsorption onto a surface with a finite number of sites.

### 3.6. Thermodynamic Analysis

The values of the thermodynamic parameters of Gibbs free energy (Δ*G*), change in enthalpy (Δ*H*), and change in entropy (Δ*S*) were calculated by using the Langmuir isotherm constant *K_L_* at different temperatures (Table 4). The following expression can be used to associate the Langmuir adsorption constant with the free energy of adsorption Δ*G* [31,42,43]:∆*G* = −*RT ln K_L_*(3)
where *T* is temperature (Kelvin), *R* is the gas constant (8.314 · 10^−3^ kJ/mol·K), and *K_L_* is the equilibrium constant obtained from the Langmuir isotherm model.

Enthalpy and entropy changes are also related to the Langmuir equilibrium constant by the following equation [3,10,42]:(4)$$\ln\;K_{L}\;=\;\frac{∆S}{R}\;-\;\frac{∆\;H}{RT}$$

The values of enthalpy change (Δ*H*) and entropy change (Δ*S*) were calculated from the slope and intercept of the plot ln*K_L_* versus 1000/T (Table 4 and Table 5).

The values of Gibbs free energy (Δ*G* > 0) indicate that the adsorption process is non-spontaneous and requires a small amount of energy; the adsorption process occurs naturally. The negative value of the standard enthalpy change Δ*H* shows that the adsorption process is exothermic. Negative values of Δ*S* represent a stable arrangement of nitrate ions on the adsorbent surface [16,42].

The value of enthalpy is also an indicator of the adsorption mechanism; it suggests an adsorption mechanism by physio-sorption through van der Waals forces if the value of enthalpy is less than 20 kJ mol^−1^, or an electrostatic type of forces between a pollutant and the adsorbent if it is between 20 and 80 kJ mol^−1^ [43].

A nitrate is a monovalent anion, and this makes the positively charged surface be a possible adsorption site that can adsorb nitrate ions through electrostatic attraction. The adsorption of nitrate ions onto adsorbents containing surface functional groups, such as the hydroxyl, carboxyl and amine groups, may relate to the electrostatic attraction, and these adsorbents become protonated at strongly acidic conditions [44].

#### Environmental Significance

Model solutions are generally used to investigate the applicability of an adsorbent, particularly where inexpensive (non-conventional) adsorbents are involved. The experimental results of this study demonstrated that spent black tea (without a chemical or thermal treatment) and thermally treated spent black tea represent abundant sources to prepare efficient and stable adsorbents, which can remove nitrate ions from aqueous solutions, and these sorbents can be easily regenerated.

### 3.7. Comparison of the Maximum Adsorption Capacity of Various Bio-Adsorbents for Nitrates

In the present study, the adsorption capacity of spent black tea and thermally treated black tea was compared with other natural or synthetic adsorbents used in the nitrate adsorption process, and the results are reported in Table 6.

## 4. Conclusions

In recent years, green chemistry strategies have been proposed for the utilization of plant waste, such as energy recovery, value-added products, or use as adsorbents. Tea waste can be carefully processed and converted into efficient biosorbent materials, and its disposal is not without environmental side effects.

The results of this study suggest that spent black tea (UBT) and thermally treated black tea (UBT-TT) could be effectively used as bio-adsorbents for the removal of nitrate ions from synthetic wastewater. These materials were characterized by different analytical and spectroscopic methods.

The adsorption isotherms were simulated by the Langmuir, Freundlich, and Temkin models.

The maximum adsorption capacities (q_e_; mg/g) for UBT and UBT-TT were 59.44 mg/g and 61.425 mg/g, respectively. The data obtained from this study were best fitted to the Freundlich adsorption isotherm applied to the equilibrium (the values were R^2^ = 0.9431 for UBT and R^2^ = 0.9414 for UBT-TT), assuming the multi-layer adsorption onto a surface with a finite number of sites. The Freundlich isotherm model could explain the adsorption mechanism. However, it was observed from the thermodynamics and adsorption isotherm results that a physical process also occurred in the UBT and UBT-TT adsorption process.

The current study indicates that spent black tea could be considered as an efficient biowaste and cost-effective material for the removal of nitrates from aqueous solutions.

In further research we propose to study the influence of competitive ions in wastewater and to determine a cost-benefit analysis to assess the implementation of these adsorbents on a large scale. For example, tea waste could also be utilized to make a bio-organic fertilizer that can be used in farming, because plant growth is aided by organic fertilizers which improve both the soil structure and the crop yields.

## Figures and Tables

**Figure 1 materials-16-04285-f001:**
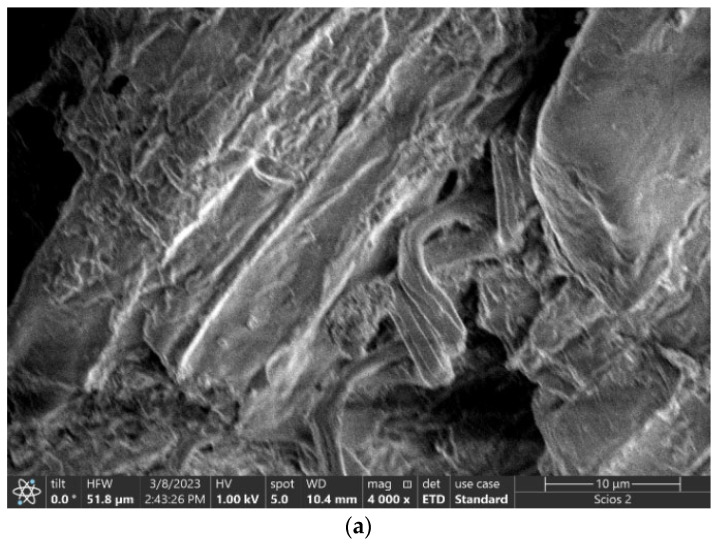
SEM images of all the adsorbent samples: (**a**) UBT before adsorption; (**b**) UBT after nitrate adsorption; (**c**) UBT-TT before adsorption; and (**d**) UBT-TT after nitrate adsorption.

**Figure 2 materials-16-04285-f002:**
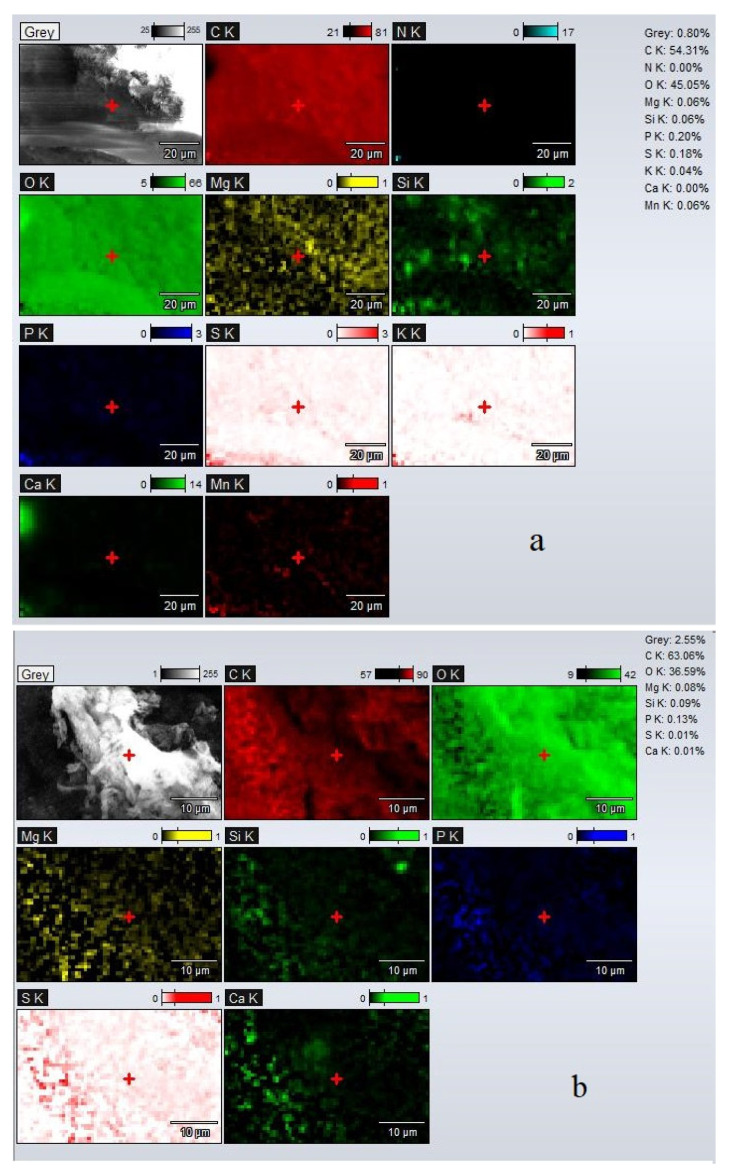
The EDX map distribution of the main elements obtained for adsorbents: (**a**) UBT before adsorption; (**b**) UBT-TT before adsorption.

**Figure 3 materials-16-04285-f003:**
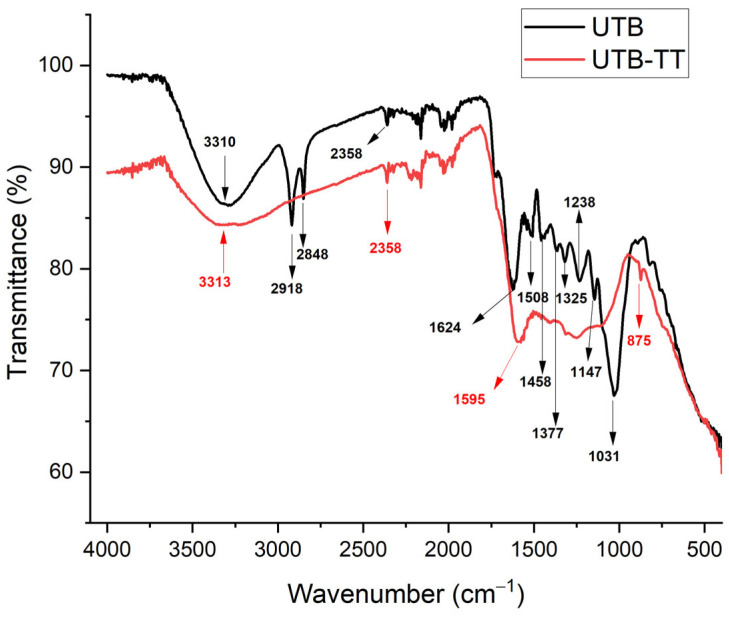
FT-IR vibrational spectra of sorbents (UBT and UBT-TT) before nitrate adsorption.

**Figure 4 materials-16-04285-f004:**
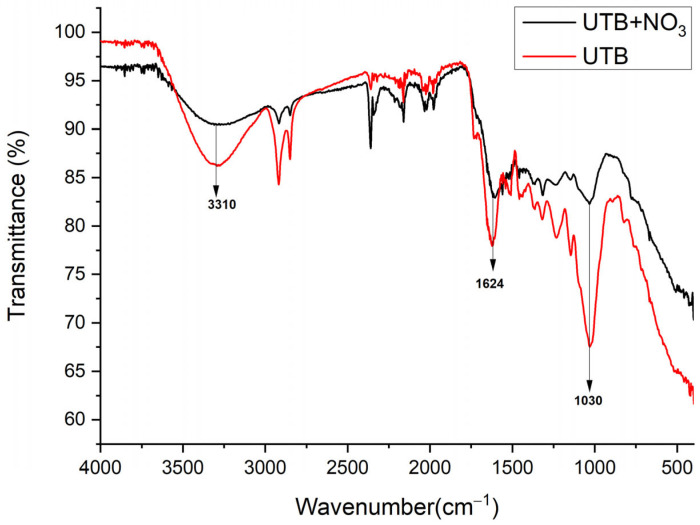
FT-IR vibrational spectra of UTB sorbent before and after nitrate adsorption.

**Figure 5 materials-16-04285-f005:**
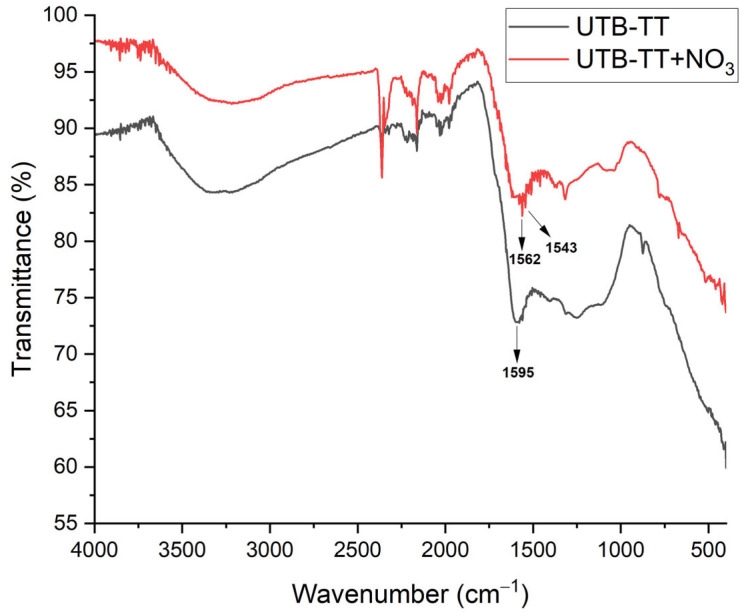
FT-IR vibrational spectra of UBT-TT sorbents before and after nitrate adsorption.

**Figure 6 materials-16-04285-f006:**
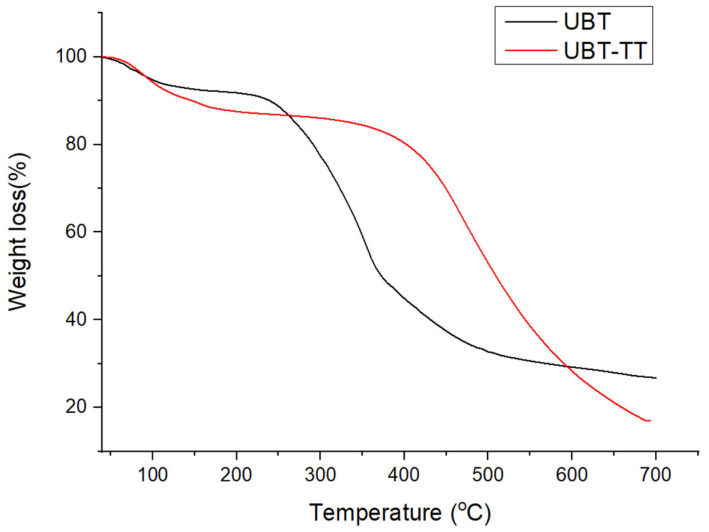
TGA curves of adsorbent samples before nitrate adsorption.

**Figure 7 materials-16-04285-f007:**
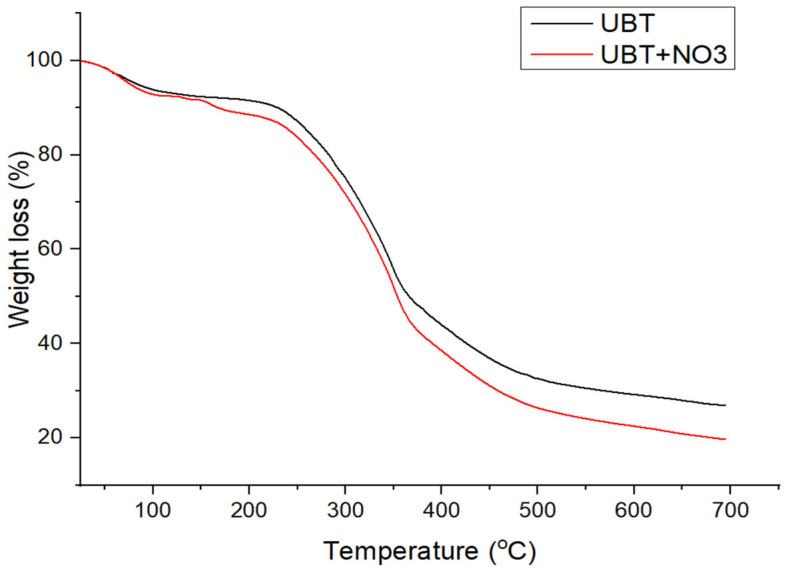
The TGA curves of UBT adsorbent samples before and after nitrate adsorption.

**Figure 8 materials-16-04285-f008:**
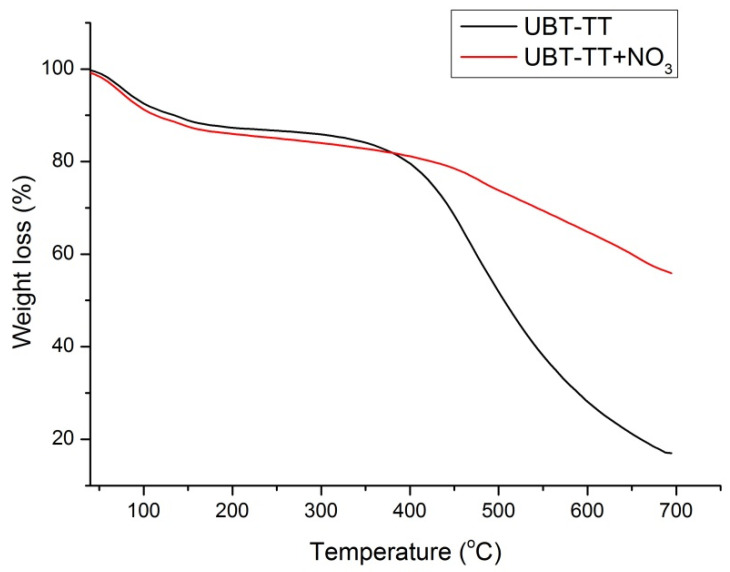
The TGA curves of UBT-TT adsorbent samples before and after nitrate adsorption.

**Figure 9 materials-16-04285-f009:**
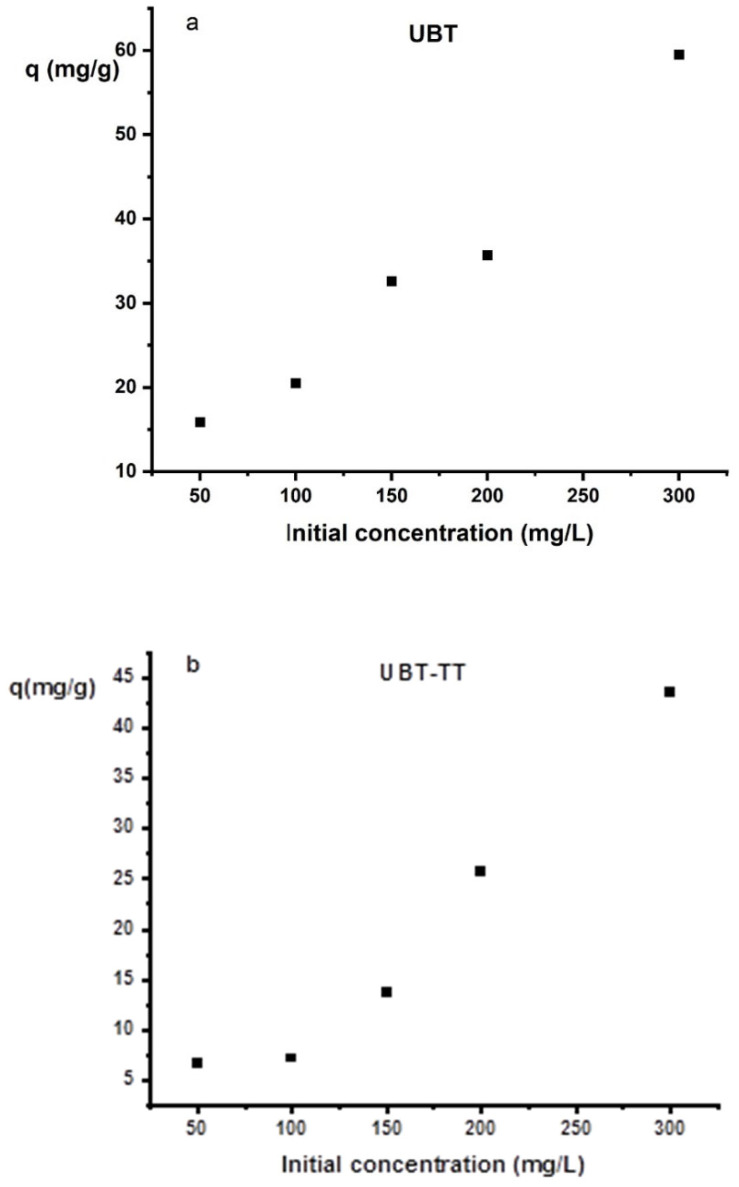
The effect of the initial concentration on nitrate adsorption on the sorbents: (**a**) UBT adsorbent; (**b**) UBT-TT adsorbent.

**Figure 10 materials-16-04285-f010:**
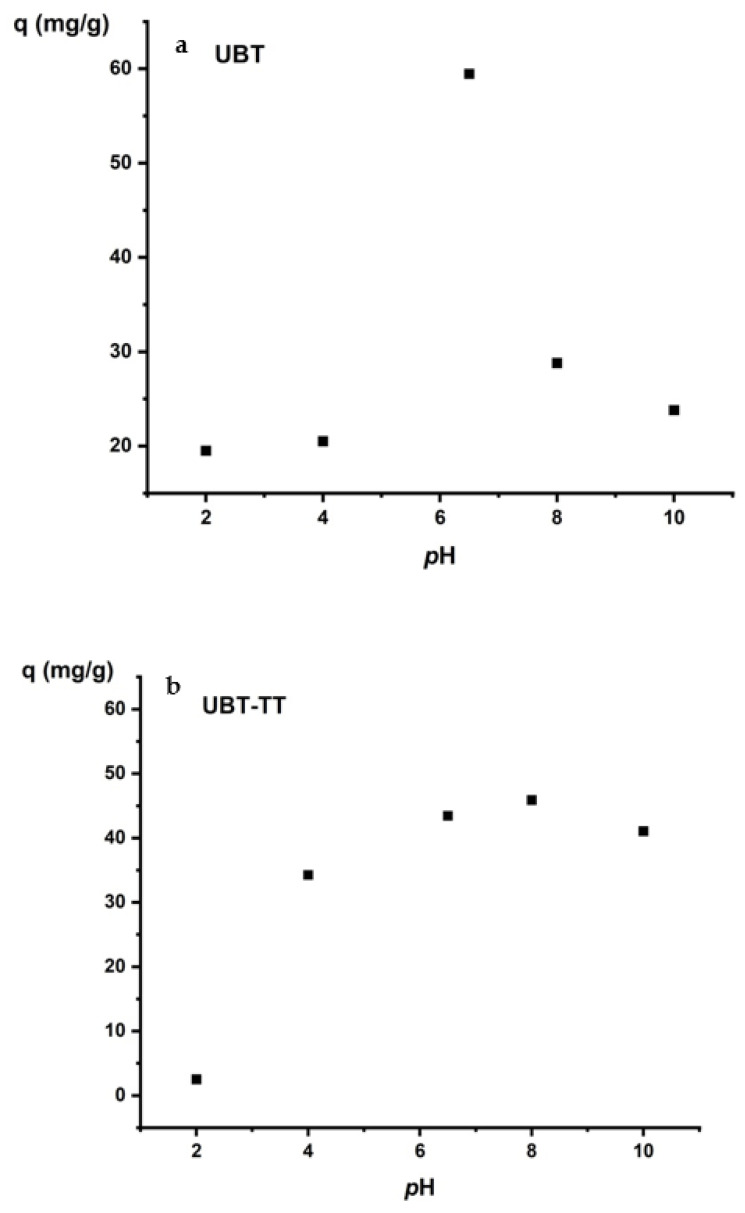
The effect of initial pH on the adsorption of nitrate on the: (**a**) UBT, and (**b**) UBT-TT.

**Figure 11 materials-16-04285-f011:**
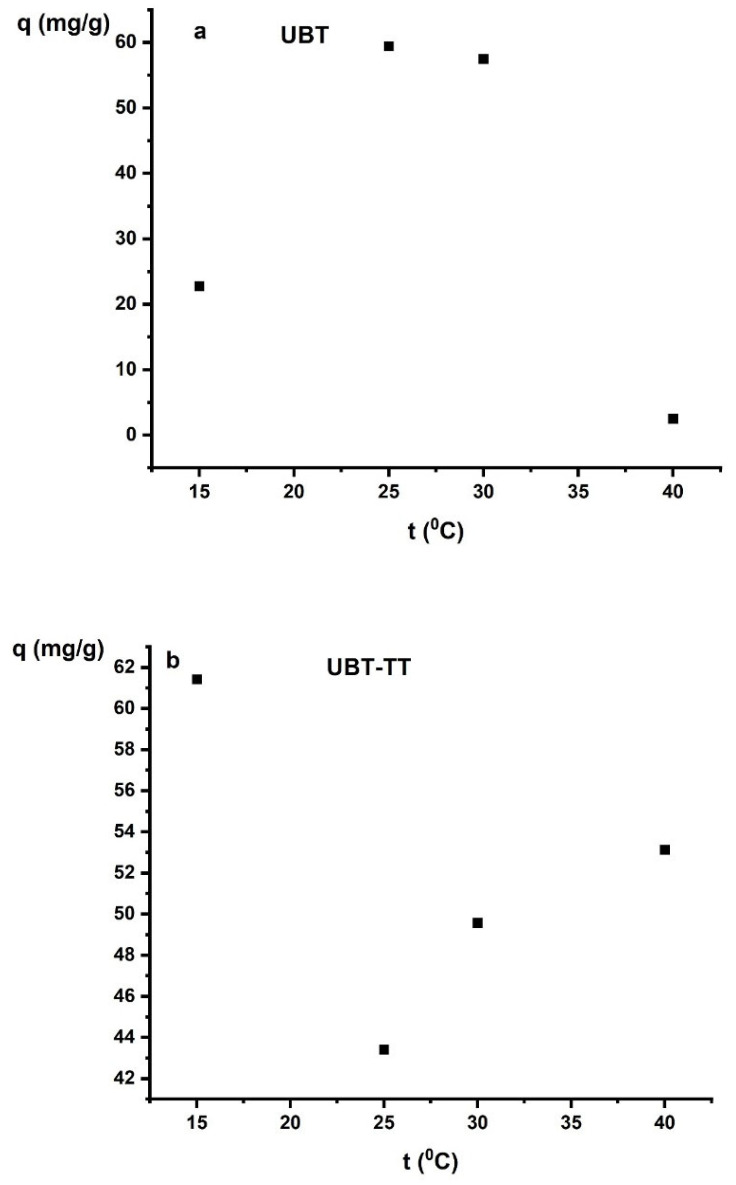
The effect of temperature on the adsorption of nitrate on the: (**a**) UBT and (**b**) UBT-TT.

**Figure 12 materials-16-04285-f012:**
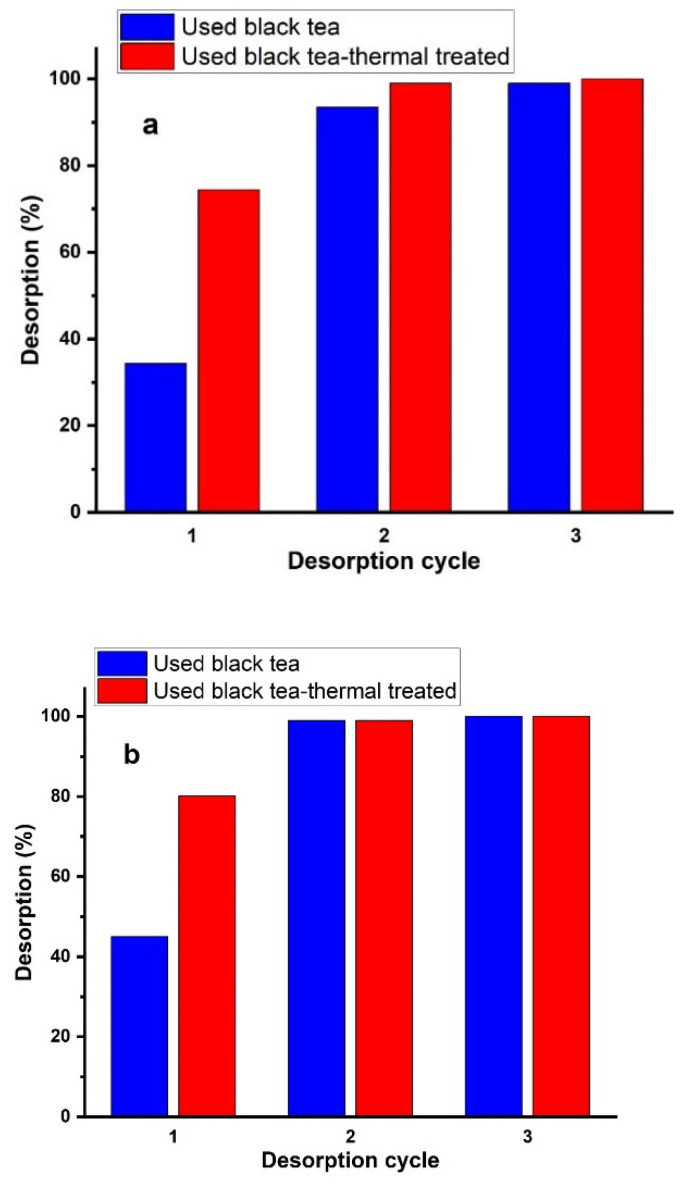
Elution of NO_3_^−^ from the adsorbents using (**a**) HCl, (**b**) NaOH, or (**c**) Na_2_CO_3_ solutions at a concentration of 0.1 mol/L in three sorption–desorption cycles.

**Figure 13 materials-16-04285-f013:**
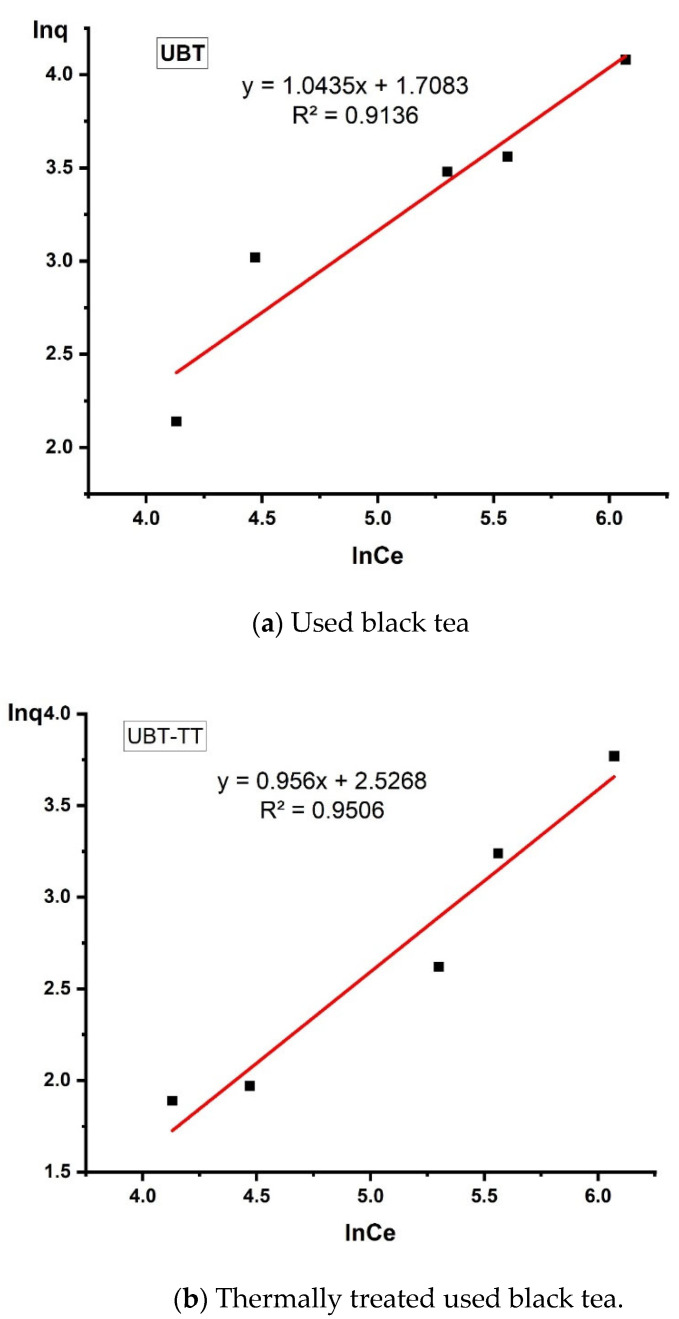
The linearized Freundlich adsorption isotherm for the adsorption of nitrate in the sorbents.

**Figure 14 materials-16-04285-f014:**
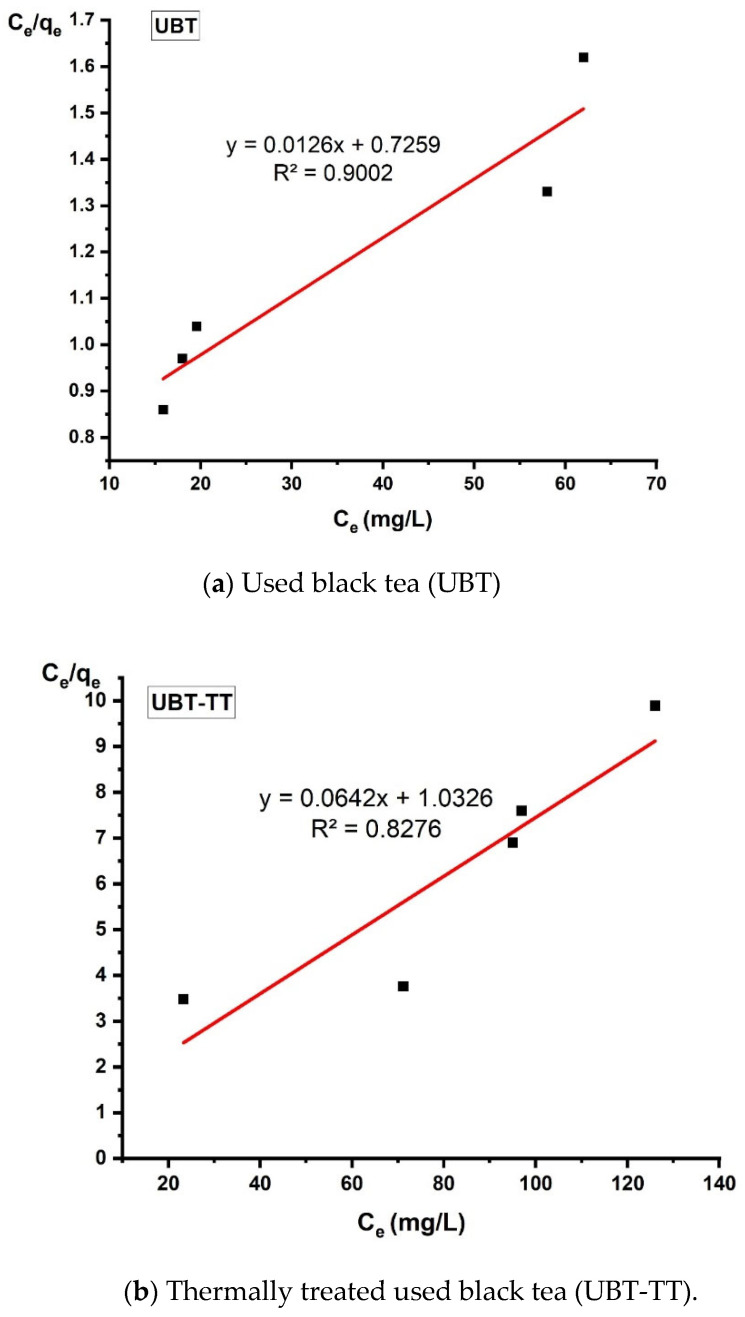
The linearized Langmuir adsorption isotherm for the adsorption of nitrate in the sorbents.

**Figure 15 materials-16-04285-f015:**
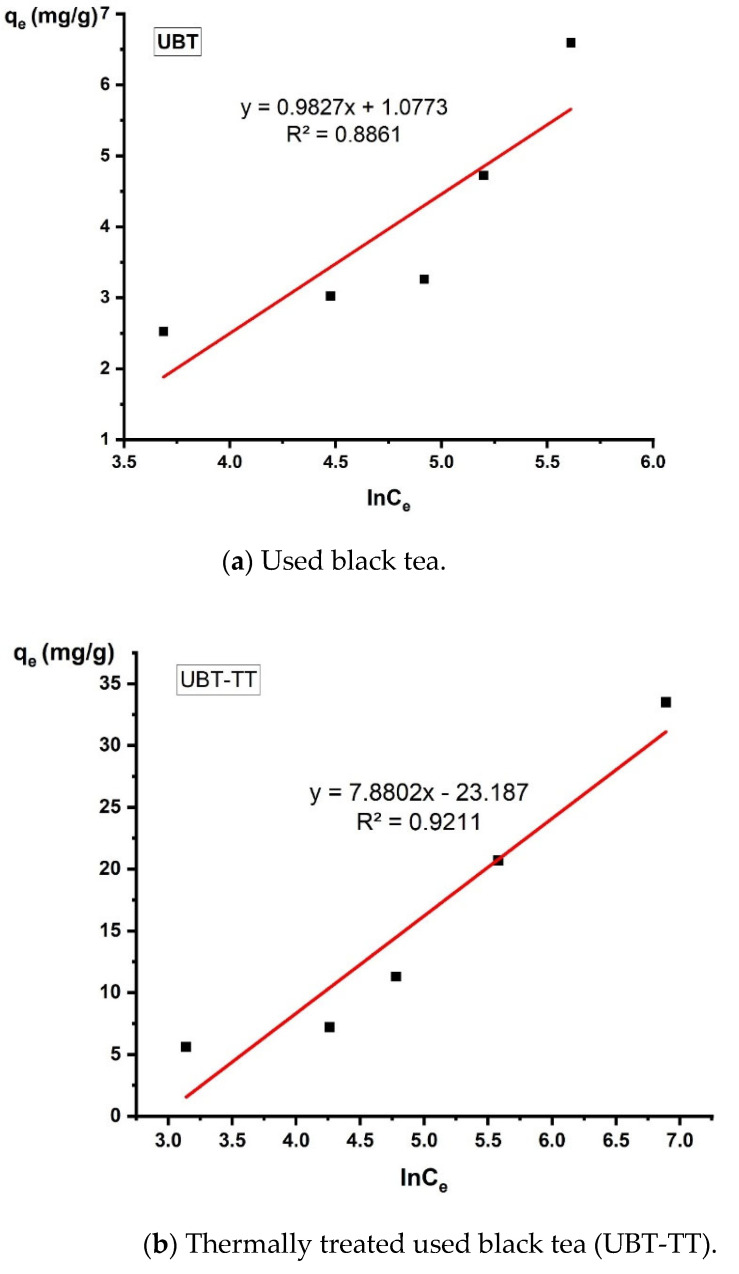
The linearized Temkin adsorption isotherm for the adsorption of nitrate on the sorbents.

**Figure 16 materials-16-04285-f016:**
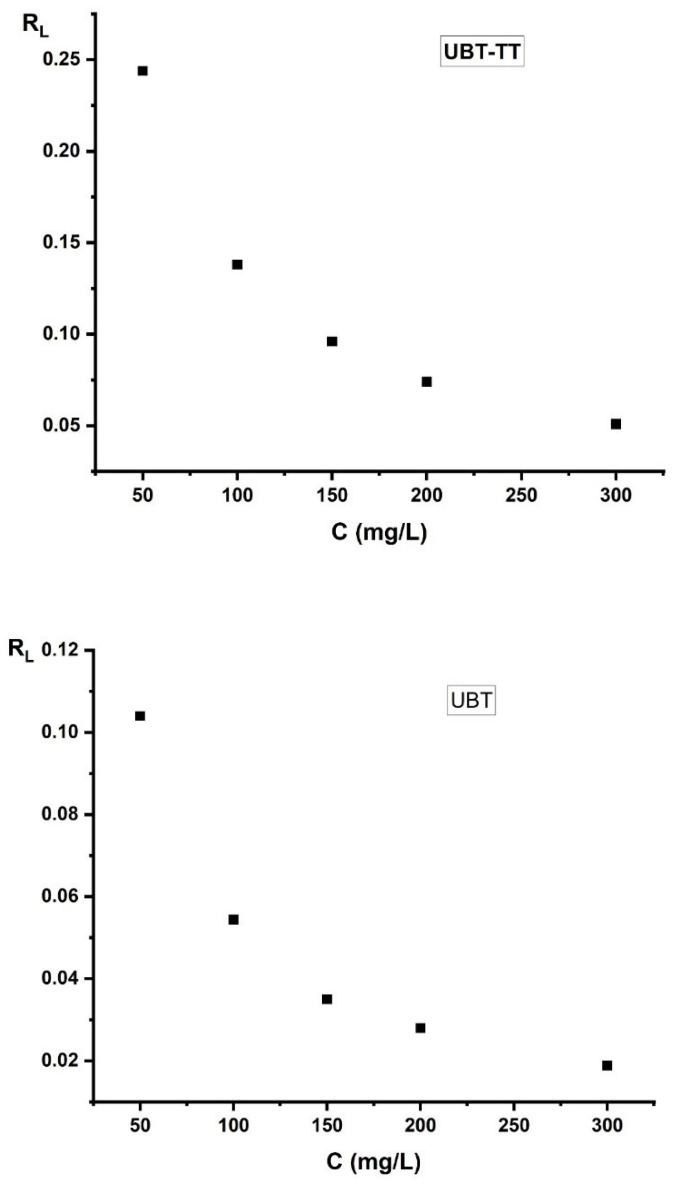
The value of separation factor *R*_L_ for the sorption of nitrate by using the UBT and UBT-TT adsorbents.

**Table 1 materials-16-04285-t001:** The equations of the Langmuir, Freundlich and Temkin isotherm models [3,10,18,28].

Langmuir	*q*_e_ = $\frac{\;q_{m}\;\cdot \;K_{L}\;\cdot \;C_{e}\;}{1\;+\;K_{L}\cdot \;C_{e}\;}$ Langmuir linearized form: $\frac{C_{e}}{\;q_{e}\;}$=$\;\frac{C_{e}}{\;q_{m}\;}$ + $\frac{1}{\;q_{m}\;\cdot \;K_{L}\;\;}$	*q_e_* = adsorption capacity determined at equilibrium (mg·g^−1^); q_m_ = maximum adsorption capacity (mg·g^−1^); *K_L_* = Langmuir constant (L·mg^−1^)
Freundlich	*q_e_* = *K_F_* · C*_e_* ^1/*n*^ Freundlich linearized form: *ln q_e_ = ln K_F_ +* $\frac{1}{n}$*ln* C*_e_*	*K_F_* = adsorption capacity; n = intensity of adsorption; *q*_e_ (mg/g) = the equilibrium sorption concentration of nitrate per gram of adsorbent; C_e_(mg/L) = the concentration of the solute in solution at equilibrium
Temkin	$q=\frac{RT}{b}\ln(K_{T}$·$C_{e}$)	*B_T_* (heat of adsorption in J·mol^−1^) = R·T/*b_T_*; *A_T_* = equilibrium-binding constant of the Temkin isotherm in L·g^−1^; *b_T_* = the Temkin isotherm constant; R = universal gas constant (8.314 J·mol^−1^ K^−1^); T = temperature (298 K)

**Table 2 materials-16-04285-t002:** The characteristics of the Langmuir adsorption isotherms [3,10,31].

Separation Factor (*R*_L_)	Type of Isotherms
*R*_L_ > 1	Unfavorable
*R*_L_ = 1	Linear
0 > *R*_L_ < 1	Favorable
*R*_L_ = 0	Irreversible

**Table 3 materials-16-04285-t003:** The parameters of the Freundlich, Langmuir and Temkin adsorption isotherm models for nitrate removal.

Adsorbent	Langmuir Model	Freundlich Model	Temkin Model
UBT	R^2^ = 0.9002 q_m_ = 7.9365 mg/g *K_L_* = 0.17357 L/mg Δ*G* = 4.3386 kJ/mol *R*_L_ = 0.0188 (*C_i_* = 300 mg/L)	R^2^ = 0.9136 *K_F_* = 2.839 mg/g 1/*n*_F_ = 0.5853	R^2^ = 0.8861 *K_T_* = 2.9921 L/g *b_T_* = 2.456 kJ/mol
UBT-TT	R^2^ = 0.8276 q_m_ = 15.5763 mg/g *K_L_* = 0.06217 L/mg Δ*G* = 6.882 kJ/mol *R*_L_ = 0.05 (*C_i_* = 300 mg/L)	R^2^ = 0.9506 *K_F_* = 2.601 mg/g 1/*n*_F_ = 0.3957	R^2^ = 0.9211 *K_T_* = 0.0527 L/g *b_T_* = 0.3063 kJ/mol

**Table 4 materials-16-04285-t004:** The thermodynamic parameters at various temperatures for the adsorption of nitrate onto spent black tea adsorbent (UBT) were an adsorbent dose of 0.2 g, a contact time of 90 min, and a pH of 6.5; q_max UBT_ = 59.44 mg/g.

*T* (K)	Δ*G* (kJ/mol)	Δ*S* (J/mol·K)	Δ*H* (kJ/mol)
288	4.21	−0.57	−0.39
293	4.59		
298	4.92		
313	5.24		

**Table 5 materials-16-04285-t005:** The thermodynamic parameters at various temperatures for the adsorption of nitrate onto spent black tea adsorbent (UBT-TT) were an adsorbent dose of 0.2 g, a contact time of 90 min, a pH of 6.5; and a q_max UBT_ = 61.425 mg/g.

*T* (K)	Δ*G* (kJ/mol)	Δ*S* (J/mol·K)	Δ*H* (kJ/mol)
288	2.58	−0.73	−0.51
293	2.79		
298	2.93		
313	3.36		

**Table 6 materials-16-04285-t006:** Comparison of the maximum adsorption capacity of various adsorbents towards NO_3_^−^ removal.

Adsorbent	q_e_ (mg/g)	Reference
Chitosan-zeolite (Ch-Z)	45.8	[28]
Chitosan-Fe(III)–Al(III)	8.58	[45]
Silica nanoparticles	14.2	[12]
Modified hazelnut shells	26.51	[19]
Amine-grafted corn cob and coconut copra	59	[46]
Modified grape seeds	27.47	[47]
Quaternized pine sawdust	29.5	[48]

## Data Availability

Data can be requested to the authors.

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
