# Peer review of "Black Tea Waste as Green Adsorbent for Nitrate Removal from Aqueous Solutions"

_materials, 2023, doi:10.3390/ma16124285_

Round 1

Reviewer 1 Report

A positive check with real time samples (used tea wastes and nitrate rich wastewater) would have better interpreted the results

Author Response

Point 1: A positive check with real time samples (used tea wastes and nitrate rich wastewater) would have better interpreted the results.

Response 1: Model solutions are usually used to study the applicability of an adsorbent, especially when low-cost (unconventional) adsorbents are involved. The composition of real wastewater is much more complex and has many contaminants that can affect the adsorption process. We purpose in a further research to study the influence of competitive ions in real wastewater and to determine a cost-benefit analysis to assess the implementation of these adsorbents on a large scale.

Reviewer 2 Report

Manuscript ID: materials-2362012

Title: Black tea waste as green adsorbent for nitrate removal from aqueous solutions

Major revision:

1.      In Abstract add numerical data from results.

2.      In introduction line 53 to 65. Discuss health effects due to nitrates in drinking water for reference, : https://doi.org/10.1080/03067319.2020.1846732.

3.      More adsorbent types should be discussed in introduction section to increase the novelty and betterreadership like Surfaces and Interfaces 34 (2022) 102324, . Also other method for pollutants removal should also be discussed and advantage of using adsorbent technique should be explored. For reference Chemical Physics Letters 805 (2022) 139939

4.      In most of the places references are cited in bulk. This practice should be avoided and only specific reference should be cited. Also intense repetition of some references should be avoided. Should be corrected in revised manuscript

5.      What is the novelty of this research? There are many reported researches in which tea waste is used for the treatment purpose.

6.      % purity /grade of chemicals used should be presented

7.      BET analysis of  UBT-TT should be carried out

8.      XRD analysis should be done forstructure analysis

9.      In table 5 add qmax of this study i.e. spent black tea (UBT) and thermally treated black tea (UBT-TT). also added experimental conditions in table 5 like adsorbent dose, contact time and pH etc

10.  Reusability of bio-adsorbent should be performed.

11.  Proposed mechanism of adsorbent should be discussed in detail

12.  There are so many typo grammatical errors in whole manuscript, should be revised by some native speaker and formatting should be checked.

There are so many typo, grammatical errors in whole manuscript, should be revised and formatting should be checked.

Author Response

Point 1: In Abstract add numerical data from results

Response 1: Data were completed.

Point 2: In introduction line 53 to 65. Discuss health effects due to nitrates in drinking water for reference, : https://doi.org/10.1080/03067319.2020.1846732.

Response 2.  We discussed more detailed health effects due to nitrates in drinking water. This reference was added.

Point 3: More adsorbent types should be discussed in introduction section to increase the novelty and betterreadership like Surfaces and Interfaces 34 (2022) 102324, . Also other method for pollutants removal should also be discussed and advantage of using adsorbent technique should be explored. For reference Chemical Physics Letters 805 (2022) 139939

Response 3: More adsorbent types were presented in Introduction section. These references were added.

Point 4: In most of the places references are cited in bulk. This practice should be avoided and only specific reference should be cited. Also intense repetition of some references should be avoided. Should be corrected in revised manuscript

Response 4: It was corrected in revised manuscript

Point 5: What is the novelty of this research? There are many reported researches in which tea waste is used for the treatment purpose

Response 5: Vegetable wastes have recently been the subject of green chemistry strategies for usage, including the recovery of energy or value-added products or the use as adsorbent materials. Tea waste can be converted to effective bio-sorbent materials, without environmental side effects. The current study shows that black tea waste can be employed as an efficient adsorbent for removing nitrate ions from aqueous solutions. Numerous research studies presented the use of black tea waste for the removal of other pollutants: dyes or heavy metals.

Point 6: % purity /grade of chemicals used should be presented

Response 6: All chemicals used in this study were of analytical grade. It was mentioned at section: Materials and methods; % purity /grade of chemicals used was presented. (Merk Millipore with 99% purity).

Point 7: BET analysis of  UBT-TT should be carried out

Response 7: Our department does not have the necessary equipment for additional structural analysis. The analysis time at another institution will be longer than 10 days and this will also generate additional costs.

Point 8: XRD analysis should be done for structure analysis

Response 8: The adsorbent materials are amorphous. They were characterized by appropriate methods (SEM, EDS, FTIR).

Point 9: In table 5 add qmax of this study i.e. spent black tea (UBT) and thermally treated black tea (UBT-TT). also added experimental conditions in table 5 like adsorbent dose, contact time and pH etc

Response 9: These values were added in the revised manuscript

Point 10: Reusability of bio-adsorbent should be performed

Response 10: A general study of the reusability of bio-adsorbent was performed and it was presented (section 3.4. Regeneration experiments).We purpose in a further research to determine a cost-benefit analysis to assess the implementation of these adsorbents on a large scale. For example, tea waste could be also utilized to make bio-organic fertilizer that can be used in farming, because plant growth is aided by organic fertilizers, which improve both soil structure and crop yields.

Point 11: Proposed mechanism of adsorbent should be discussed in detail

Response 11: Proposed mechanism of adsorbent was presented in the revised manuscript

Point 12: There are so many typo grammatical errors in whole manuscript, should be revised by some native speaker and formatting should be checked

Response 12: Manuscript was revised by a native speaker and formatting was checked. 

Reviewer 3 Report

In this study, the authors used black tea waste as green adsorbent for nitrate removal from aqueous solutions. The topic is interesting and valuable, but this paper needs to revise carefully in many places before it can be considered in a high-impact journal like Materials. Hope below comments will be able to help to further improve the paper:

1. Abstract, the expression of the results needs to be clearer.

2. The introduction should be re-considered and the unnecessary information could be deleted and the explanation of the novelty should be supplemented in detail.

3. It is recommended to extend the discussion and comparison of the study with other similar published work.

4. The recovery of tea residue and the preparation of adsorbent materials need economic calculation to ensure the rationality of this study.

5. The two related literature research work should be cited: Environmental Science & Technology 2022, 56(14): 10412-10422; Environmental Science & Technology 2023, 57(11): 4591-4597.

6. English must be improved. There are some confusing sentences, wrong verb tenses, and grammatical errors in the manuscript that can be corrected with a more careful revision.

In this study, the authors used black tea waste as green adsorbent for nitrate removal from aqueous solutions. The topic is interesting and valuable, but this paper needs to revise carefully in many places before it can be considered in a high-impact journal like Materials. Hope below comments will be able to help to further improve the paper:

1. Abstract, the expression of the results needs to be clearer.

2. The introduction should be re-considered and the unnecessary information could be deleted and the explanation of the novelty should be supplemented in detail.

3. It is recommended to extend the discussion and comparison of the study with other similar published work.

4. The recovery of tea residue and the preparation of adsorbent materials need economic calculation to ensure the rationality of this study.

5. The two related literature research work should be cited: Environmental Science & Technology 2022, 56(14): 10412-10422; Environmental Science & Technology 2023, 57(11): 4591-4597.

6. English must be improved. There are some confusing sentences, wrong verb tenses, and grammatical errors in the manuscript that can be corrected with a more careful revision.

Author Response

Point 1: Abstract, the expression of the results needs to be clearer

Response 1: Data were completed.

Point 2: The introduction should be re-considered and the unnecessary information could be deleted and the explanation of the novelty should be supplemented in detail

Response 2. The introduction was re-considered (see revised manuscript)

Point 3: It is recommended to extend the discussion and comparison of the study with other similar published work.

Response 3: Data were completed.

Point 4: The recovery of tea residue and the preparation of adsorbent materials need economic calculation to ensure the rationality of this study.

Response 4: Various approaches can be used for adsorbent cost analysis.

For example, we used this economic calculation: Cost of adsorbent per gram of the adsorbate removed:

The economic viability of the adsorbent prepared can be determined by comparing the cost of the adsorbent per gram of adsorbate removed with that of other well-known adsorbents, assuming the operational and regeneration costs are similar. The cost of an adsorbent per gram of adsorbate can be used to determine the cost of the adsorption operation as shown in this Eq. (Bajić et al., 2016):

where: the adsorption capacity is the amount of adsorbate removed per adsorbent used in mg/g. This can be also scaled up for large-scale industrial applications.

Summary of adsorbent types, precursors and processing techniques:

UBT biosorbent: precursors-used black tea; production: washing-drying-grinding-sieving.

UBT-TT biosorbent: precursors-used black tea; production: washing-drying-carbonization-sieving.

We simulated an economic calculation for these two adsorbents: UBT and UBT-TT and we obtained these results:

UBT: 0.04$/g (cost per treatment); UBT-TT: 0.18 $/g (cost per treatment).

Point 5: The two related literature research work should be cited: Environmental Science & Technology 2022, 56(14): 10412-10422; Environmental Science & Technology 2023, 57(11): 4591-4597.

Response 5: The Introduction part was modified. These two related literature research work were cited.

Point 6: English must be improved. There are some confusing sentences, wrong verb tenses, and grammatical errors in the manuscript that can be corrected with a more careful revision.

Response 6: Manuscript was revised by a native speaker and formatting was also checked

Reviewer 4 Report

This manuscript reports the study of the removal of nitrate from contaminated water using black tea waste. The authors investigated the possibility to use black tea waste as an adsorbent of nitrate and used untreated and thermally treated tea waste. The described study does not fit with the scopes of Materials MDPI and should be submitted in a journal more specialized in the field of water remediation and water treatment. The part related to Materials characterization and study is short. I would not recommend this manuscript for publication, the manuscript must be improved and some parts must be corrected. The wording is also incorrect in some parts (i.e. adsorbed activated carbon instead of exhausted activated carbon...). Here are the detailed comments:

-          The authors should check the exact concentration of the synthetic solution they prepared. If they used 3.6090 g of KNO3 to prepare the solution, the concentration is not 1g/L.

-          For the preparation of the tea waste that will be used for nitrate removal, the authors performed a lot of treatments that use energy and water (boiling 8 times and then dried). The authors must comment it in the manuscript. Is it cost effective and what about fresh water used for the preparation.

-          The preparation of thermally treated requested in addition, treatment at 400C and rinsing till a neutral pH is obtained. What about cost effectiveness and eco-friendliness?

-           The authors used HCl (0.1M) to adjust the pH, however later in the manuscript, the authors mention that HCl is used to regenerate the exhausted tea waste. It is an important issue in the manuscript and presented study. This fact can explain the problem in figure 9A.

-          Calibration curve to measure NO3 concentration must be provided. In addition, the calibration was done with probably a NO3 solution that is not 1g/L.

-          The term “wide field” is not correct.

-          SEM image in figure 1d, shows probable salt formation, EDS must be performed on the white particles.

-          FTIR does not show any presence of NO3. They must be visible if NO3 was adsorbed on tea waste.

-          How much is the starting mass in TGA measurement? If the mass is different for each test, you cannot compare the weight loss in different samples.

-          In the adsorption tests, the authors must add missing information (i.e. temperature in fig. 8, temperature and concentration in fig. 9, pH in fig. 10)

-          Figure 9 shows that adjusting pH with HCl affects the capacity of adsorption. Se comment above about HCl usd for regeneration.

-          The regeneration of tea waste does not appear cost-effective (i.e. 3 cycles with water and HCl use, and filtration.

-          R factor should be around 0.96-0.99. Lower R shows an inaccurate fitting. How fig. 13a can be fitted? Measures are only localized at 20 and 60 mg/L. Error bar of fitting will be very high.

-          If we consider eq. (7), RL cannot be 0, or KL must be very high, which cannot happen.

-          Page 21, the authors mention a value of nF between 1 to 10, but the values they gave are lower than 1.

-          Page 22, environmental significance the authors must consider the water consumption for the preparation of the tea waste and the regeneration.

-          Page 22, part 3.7 must be discussed and improved. It looks that the other adsorbents have better adsorption capacity.

- What about qin the conclusion?

The English must to be improved, the manuscript should be carefully read by a person proficient in English; there are numerous typos and grammatical errors that should be corrected.

Author Response

Point 1: The authors should check the exact concentration of the synthetic solution they prepared. If they used 3.6090 g of KNO3 to prepare the solution, the concentration is not 1g/L

Response 1: The exact concentration of the synthetic solution was checked (see the modification in the revised manuscript)..

Point 2: For the preparation of the tea waste that will be used for nitrate removal, the authors performed a lot of treatments that use energy and water (boiling 8 times and then dried). The authors must comment it in the manuscript. Is it cost effective and what about fresh water used for the preparation

Response 2. Various approaches can be used for adsorbent cost analysis. For example, we used this economic calculation: Cost of adsorbent per gram of the adsorbate removed:

The economic viability of the adsorbent prepared can be determined by comparing the cost of the adsorbent per gram of adsorbate removed with that of other well-known adsorbents, assuming the operational and regeneration costs are similar. The cost of an adsorbent per gram of adsorbate can be used to determine the cost of the adsorption operation as shown in this Eq. (Bajić et al., 2016):

where: the adsorption capacity is the amount of adsorbate removed per adsorbent used in mg/g. This can be also scaled up for large-scale industrial applications.

We purpose in a further research to study the influence of competitive ions in wastewater and to determine a cost-benefit analysis to assess the implementation of these adsorbents on a large scale. For example, tea waste could be also utilized to make bio-organic fertilizer that can be used in farming, because plant growth is aided by organic fertilizers, which improve both soil structure and crop yields.

Point 3: The preparation of thermally treated requested in addition, treatment at 400C and rinsing till a neutral pH is obtained. What about cost effectiveness and eco-friendliness?

Response 3: We explained this aspect above.

Point 4: The authors used HCl (0.1M) to adjust the pH, however later in the manuscript, the authors mention that HCl is used to regenerate the exhausted tea waste. It is an important issue in the manuscript and presented study. This fact can explain the problem in figure 9A.

-          Calibration curve to measure NO3- concentration must be provided. In addition, the calibration was done with probably a NO3- solution that is not 1g/L.

Response 4: The calibration was done with a solution 500mg NO3-/L. The calibration curve to measure NO3- concentration is presented below:

Point 5: The term “wide field” is not correct

Response 5: Manuscript was revised by a native speaker. Some grammatical errors in the manuscript were corrected with a more careful revision

Point 6: SEM image in figure 1d, shows probable salt formation, EDS must be performed on the white particles.

Response 6: EDS analysis was performed. Manuscript was revised and EDS analysis results were added (Fig. 2a,b).

Point 7: FTIR does not show any presence of NO3. They must be visible if NO3 was adsorbed on tea waste

Response 7: The FT-IR spectra of tea biomass before and after nitrate loading were very similar. The only significant difference was the decrease in peaks absorbance at 3310, 1624 and 1031cm-1 indicated that the -OH and C-O groups were involved in nitrate adsorption. This affirmation is sustained by the observation of Gao, Y.; Bao, S.; Zhang, L.; Zhang, L. Nitrate removal by using chitosan/zeolite molecular sieves composite at low temperature: Characterization, mechanism, and regeneration studies. Desalin Water Treat 2020, 203, 160-171.  

The same behavior was also observed for thermal treated tea biomass (UBT-TT) before and after nitrate adsorption. A slight difference was observed consisting of the splitting of the band at 1595 cm-1 into two small sharp bands at 1562 and 1543 cm-1, probably due to the N-H bending vibration of the secondary amide and the primary amine. This affirmation is sustained by the observation of Luo, S.; Li, X.; Chen, L.; Chen, J.; Wan, Y.; Liu, C. Layer-by-layer strategy for adsorption capacity fattening of endophytic bacterial biomass for highly effective removal of heavy metals. Chemical Engineering Journal 2014, 239, 312-321, doi:https://doi.org/10.1016/j.cej.2013.11.029.

Point 8: How much is the starting mass in TGA measurement? If the mass is different for each test, you cannot compare the weight loss in different samples.

Response 8: The masses used for the TGA analysis vary between 6.65 and 10.76 mg, for this reason the thermogravimetric analysis was expressed in mass loss percentages not mg.

Point 9: In the adsorption tests, the authors must add missing information (i.e. temperature in fig. 8, temperature and concentration in fig. 9, pH in fig. 10)

Response 9: These details were added in the revised manuscript.

Point 10: Figure 9 shows that adjusting pH with HCl affects the capacity of adsorption. See comment above about HCl used for regeneration

Response 10: Several attempts were performed to regenerate nitrate ions from the adsorbed support by using HCl, NaOH and Na2CO3 solutions at the concentration of 0.1 mol/L. Among them, Na2CO3 and HCl solutions had the best regeneration effect due to the enhancement of electrostatic repulsion.

Point 11: The regeneration of tea waste does not appear cost-effective (i.e. 3 cycles with water and HCl use, and filtration

Response 11: The results showed that the third cycle was 100% for HCl and Na2CO3 regeneration reagents, which demonstrated that adsorbent materials prepared from spent black tea had good reusability. The costs are not significant for this cycle.

Point 12: R factor should be around 0.96-0.99. Lower R shows an inaccurate fitting. How fig. 13a can be fitted? Measures are only localized at 20 and 60 mg/L. Error bar of fitting will be very high.

Response 12: Please consider some examples from these references:

https:// doi.org/10.3390/w14182906; https://doi.org/10.3390/w14050816;

doi: 10.2166/aqua.2022.076; https://doi.org/10.1016/j.jhazmat.2020.122441

R2 factor was not around 0.96-0.99 in these studies, but it was considered for a good modelling data fitting. These are mathematical models and the best value of R2 factor is considered. In order to find out the surface properties and affinity of the adsorbent for adsorbates, different isotherm models were applied. In this case, the correlation coefficient R2 values are used to obtain the best fit linear equation. For example, the Freundlich isotherm provided the best correlation coefficient R2 value and hence, the best fit for the experimental data.

Point 13: If we consider eq. (7), RL cannot be 0, or KL must be very high, which cannot happen

Response 13: The Langmuir parameters can be used to predict affinity between nitrate and each sorbent using a dimensionless separation factor (RL):

RL =                          (7)

Where: KL is the Langmuir constant and Ci is the initial concentration (mg/L). 

Please consider also these supplementary references:

https://doi.org/10.1007/s13201-016-0403-6; https://doi.org/10.3390/ ma15238566

https://doi.org/10.1515/tjb-2017-0333; DOI: http://dx.doi.org/ DOI:10.5714/CL.2017.22.014.

Point 14: Page 21, the authors mention a value of nF between 1 to 10, but the values they gave are lower than 1.

Response 14;: This parameter was calculated again and it was corrected, the value of 1/ nF is given in the revised form of our manuscript (Table 3).

Point 15: Page 22, environmental significance the authors must consider the water consumption for the preparation of the tea waste and the regeneration.

Response 15: The adsorbent loaded with nitrate can be used as a controlled-release fertilizer, thus reducing water consumption for its regeneration

Point 16: Page 22, part 3.7 must be discussed and improved. It looks that the other adsorbents have better adsorption capacity

Response 16: The maximum adsorption capacities (qe; mg/g) for UBT and UBT-TT were 59.44 mg/g and 61.425 mg/g respectively. Other adsorbents presented in table 5.

Point 17: What about qe in the conclusion?

Response 17: The values of qe were added in the conclusion (see revised manuscript).

Round 2

Reviewer 2 Report

Accept

Minor editing of English language and spell check required

Author Response

The manuscript has been revised. Changes are highlighted in blue.

Reviewer 3 Report

The manuscript has been improved greatly and the authors give positive responses to the comments. It can be accepted.

Author Response

Thank you for the appreciation.

Reviewer 4 Report

The authors did not reply to the following comment:

The authors used HCl (0.1M) to adjust the pH, however later in the manuscript, the authors mention that HCl is used to regenerate the exhausted tea waste. It is an important issue in the manuscript and presented study. This fact can explain the problem in figure 9A.

Added EDS part is not discussed. How Na and Mn elements can appear after thermal treatment?

I am not convinced by the presented study, and I would not recomment the publication of this study in Materials MDPI.

Author Response

Point 1: The authors used HCl (0.1M) to adjust the pH, however later in the manuscript, the authors mention that HCl is used to regenerate the exhausted tea waste. It is an important issue in the manuscript and presented study. This fact can explain the problem in figure 9A.

Response 1: Thank you very much for the indication. We completed the text with the following paragraph: „Solution pH has an important effect on nitrate adsorption. Most adsorbents lose their adsorption capacity under strongly acidic or basic conditions. The change in adsorption with pH (Figure 9) can be explained by electrostatic interactions between adsorbents and adsorbates. The lowest values of the adsorption capacity were obtained at pH 2 and 4, for UBT and UBT-TT. Adjusting the pH with HCl affects the adsorption capacity. At lower pH values the H+ ions compete with nitrate ions for the electrostatic surface charges in the system decreasing the percentage of sorption [41]. The best values of the adsorption capacity were obtained at pH 6.5 for UBT and pH 8 for UBT-TT (Fig.9 -a,b).  The highest adsorption capacity was achieved at neutral pH, mainly due to strong electrostatic interactions between the adsorption sites and nitrate anions. A lower adsorption of nitrate at alkaline pH may be due to abundant HO- ions competing with contaminants for the same sorption sites” (pages14-15, lines:409-423).

Point 2: Added EDS part is not discussed.

Response 2: We completed the text with the following paragraph:”The EDX spectra of adsorbents (UBT and UBT-TT) are shown in Figure 2. The EDX spectra of adsorbent sample UBT showed that on the surface of all the samples, carbon and oxygen were mostly present. The EDX spectra also showed the presence of other elements, such as Mg, K, Mn, Si, P, S and Mn (figure 2.a). After the thermic treatment the content of carbon increase from 54.13 to 63.06%, meanwhile the oxygen decrease from 45.05 to 36.59% due to the loss of moister and some phosphate, and sulfate groups. This affirmation is supported by the decrease in the values recorded for P and S from 0.20 to 0.13% and respectively 0.18 to 0.1% (figure 2.b). This behavior was also reported by P. Wijeyawardana et al.[34] and S. Suman et al.[35].(page7, lines 283-293. ”

Point 3: How Na and Mn elements can appear after thermal treatment?

Response 3. We apologize due to a technical error we confused the EDX images. We reanalyzed the samples and noticed that the EDX analysis for UBT was mistakenly attributed to the UBT-TT sample. The reanalysis EDX for UBT has been added.